# Comparative Analysis of Submaximal and Maximal Effort Capacities in Patients Post-COVID-19 and Individuals with Chronic Restrictive Lung Diseases

**DOI:** 10.3390/ijerph22020261

**Published:** 2025-02-12

**Authors:** Karissa Yasmim Araújo Rosa, Felipe Xavier de Melo, Fernanda Lara Fernandes Bonner Araújo Riscado, Rodrigo F. Oliveira, Deise A. A. P. Oliveira, Iransé Oliveira-Silva, Luís V. F. Oliveira, Dante Brasil Santos

**Affiliations:** 1Program in Human Movement and Rehabilitation at the Anápolis University Center, Av Universitária km 3.5, Bloco B2, sl 501, Anápolis 75083-515, GO, Brazil; karissayasminar@gmail.com (K.Y.A.R.); rodrigofranco65@gmail.com (R.F.O.); deisepyres@gmail.com (D.A.A.P.O.); iranseoliveira@hotmail.com (I.O.-S.); oliveira.lvf@gmail.com (L.V.F.O.); 2Pulmonary Unit, Hospital Universitário de Brasília, Universidade de Brasília, SGAN 604/605, Av L2 norte, Brasília 70840-901, DF, Brazil; felipe.melo@ebserh.gov.br (F.X.d.M.); febonner@hotmail.com (F.L.F.B.A.R.); 3Pulmonary and Metabolic Rehabilitation Program, Hospital Universitário de Brasília, Universidade de Brasília, SGAN 604/605, Av L2 norte, Brasília 70840-901, DF, Brazil; 4Reference Center for Neuromuscular Diseases, Hospital de Apoio de Brasília, Brasília 70684-831, DF, Brazil

**Keywords:** acute post-COVID-19 syndrome, lung diseases, restrictive, spirometry, 6 min walk test, cardiopulmonary exercise test

## Abstract

Whether impairments in submaximal and maximal effort capacities in individuals following acute COVID-19 infection resemble those found in patients with chronic pulmonary disease remains unclear. We aimed to analyze the submaximal and maximal effort capacities of patients after COVID-19 infection and those with alterations in lung mechanics similar to those observed in patients with chronic respiratory diseases. This retrospective cross-sectional observational study paired a group of post-COVID-19 individuals with another group of patients with chronic respiratory disease, using spirometric patterns similar to those observed post-COVID-19. Data from Spirometry, 6 min walk test (6-MWT), and cardiopulmonary exercise test (CPET) variables were compared, and correlations between spirometric variables and 6-WT/CPET were examined. The final sample comprised 20 patients, including 10 post-COVID-19 patients with a restrictive lung disease (RLD) pattern identified using spirometry and 10 patients with RLD. Both groups presented similar patterns of the analyzed variables, with significant correlations observed between forced vital capacity (FVC) the distance and speed achieved during the 6-MWT, and a negative correlation between FVC and V’ E max. The degree of restriction in the overall sample influenced the covered distance and speed during the 6-MWT as well as the maximum minute ventilation during maximal effort.

## 1. Introduction

In recent years, healthcare professionals and the general population have faced significant challenges due to the SARS-CoV-2 pandemic [1]. COVID-19, caused by direct contact with the SARS-CoV-2 virus, greatly strains healthcare systems worldwide. Symptoms of infection include fever, dyspnea, cough, proteinuria, and skeletal muscle wasting, reflecting its systemic nature [2]. Severe and critical cases often require supplemental oxygen and mechanical ventilation, and patients are at an increased risk of long-term sequelae [3].

A post-COVID-19 syndrome has been recognized, characterized by persistent symptoms lasting > 4 weeks after the initial infection, which cannot be attributed to other pathophysiological causes [4]. The most common symptoms are impaired functional capacity and exertional dyspnea [5], which necessitates intervention through a rehabilitation program aimed at restoring overall function.

Pulmonary rehabilitation programs, typically designed for patients with chronic lung disease, rely on comprehensive patient evaluations [6]. This evaluation might encompass the assessment of submaximal effort capacity using the 6-min walk test (6-MWT), which can be improved by analyzing maximal effort capacity using the cardiopulmonary exercise test (CPET). These tests help gauge the extent of functional impairment, determine the need for supplemental oxygen during exercise, and identify the most affected physiological systems (respiratory, cardiovascular, or musculoskeletal) for targeted exercise interventions.

Post-COVID-19 patients frequently exhibit restrictive spirometric patterns, with reduced forced vital capacity (FVC) and forced expiratory volume in the first second (FEV_1_) [7,8]. Furthermore, a systematic review and meta-analysis [9] have identified the restrictive lung pattern as the most prevalent mechanical pulmonary impairment observed in patients with post-COVID-19 infection. Similarly, impairments in functional capacity and exertional ability have been documented in patients with restrictive lung disease (RLD) [10,11]. Persistent functional impairments have been documented in post-COVID-19 patients over extended follow-up periods, with some individuals exhibiting reduced exercise capacity, dyspnea, and pulmonary function abnormalities post-infection [12]. While some patients show gradual improvement, others experience prolonged deficits, particularly those with severe diseases requiring mechanical ventilation [13]. This suggests that while post-COVID-19 impairments may be transient in some patients, others may develop long-term functional limitations comparable to those seen in patients with chronic pulmonary diseases. Therefore, given the prevalence of pulmonary restriction post-COVID-19, even if considered a transient condition, it is reasonable to assume that derangements in submaximal and maximal effort capacities may be comparable to those observed in patients with RLD. Furthermore, it is crucial to understand functional impairment patterns, which might not be properly evaluated using resting physiological evaluations, and identify the factors contributing to pulmonary-related exercise limitations.

To better understand these patterns and identify the factors, we analyzed submaximal and maximal effort variables in patients diagnosed with post-COVID-19 infection and in another group of patients with chronic respiratory disease, paired with pulmonary function patterns. Because the performance of the groups was similar in terms of pulmonary function, we analyzed it as a unique sample. We looked for correlations between the pulmonary function test results and those of the 6-MWT and CPET.

## 2. Materials and Methods

### 2.1. Study Design

We conducted a retrospective observational cross-sectional study using data from the electronic medical records of a pulmonary rehabilitation program within a hospital setting and data collected from participants evaluated during the study period. This study was approved by the Research Ethics Committee of Unievangélica (Anápolis, GO, Brazil) (approval number: 4.296.707) and was part of a larger study registered at ClinicalTrials.gov—ID: COVID-19 PULMONARY REHAB NCT 04982042).

### 2.2. Data Collection and Participants

This study included adult participants of both sexes, aged 18–70 years, who had medical indications and received approval for follow-up in a pulmonary rehabilitation program following acute SARS-CoV-2 infection between August 2020 and January 2021. Weeks after the initial infection, these patients still exhibited major pulmonary convalescence symptoms, such as dyspnea, exertional breathing, or fatigue upon exertion, as the main causes of pulmonary rehabilitation program follow-up. Individuals with other indications for pulmonary rehabilitation; those who were clinically unstable, or those with physical dependence and/or neurological, orthopedic, or psychiatric conditions that could impede participation in the protocol were excluded. Additionally, individuals with chronic respiratory diseases referred to the pulmonary rehabilitation program between February 2012 and February 2018 were retrospectively identified to collect data from patients who presented with spirometric patterns similar to those of the post-COVID-19 infection group. The exclusion criteria for the post-COVID-19 group included a diagnosis of any associated lung disease, cancer history, refusal to participate in the rehabilitation program, clinical instability, age <18 or >70 years, inability to understand simple commands, history of smoking, absence of a cancer diagnosis, and lack of familiarity with cycle ergometers. The exclusion criteria for patients with similar spirometric patterns as that of the post-COVID-19 infection group, patients were excluded if they presented mixed spirometric patterns (i.e., obstructive and restrictive), if they presented a history of smoking, diagnosis of any cancer anywhere, out previous respiratory exacerbation for, at least 6 months.

Initially, the patients underwent a pulmonary function test to determine whether any alterations in spirometry were present.

### 2.3. Pulmonary Function Test and Evaluation of Submaximal and Maximal Effort Capacity

All patients in the post-COVID-19 group were initially evaluated between 1–2 months after hospital discharge. The spirometry, 6-MWT, and CPET assessments were conducted by the same professionals (a lung-function respiratory technician, physiotherapist, and pulmonologist, respectively). The initial evaluation included collecting demographic data (weight, height, and age), information about the severity of SARS-CoV-2 infection (use of supplemental oxygen and noninvasive or invasive mechanical ventilation), and coexisting comorbidities.

#### 2.3.1. Spirometry Test

A pulmonary function test using spirometry was performed to determine the patterns of lung mechanics. This test was conducted using a flux spirometer (KoKo spirometer, KoKo PFT System, nSpire Health, Inc., Longmont, CO, USA), from which forced vital capacity (FVC), forced expiratory volume in one second (FEV_1_), and the relation between FEV_1_/FVC were obtained. All the parameters were obtained as absolute and predicted values and/or relative measures. The tests were performed, and at least three acceptable and reproducible forced expiratory maneuvers were performed according to established criteria. Reference values were obtained from the Brazilian spirometry consensus [14]. Both groups underwent the 6-MWT and CPET on separate days.

#### 2.3.2. 6-MWT

Functional capacity or submaximal effort capacity was assessed using the 6-MWT in a 30 m unobstructed corridor [15]. Briefly, patients were encouraged to walk as fast as possible for 6 min, allowing them to stop if necessary. The distance walked during the 6-MWT (D6-MWT), peripheral oxygen saturation (SpO_2_), and heart rate (HR) were measured at the beginning and end of the test. Additionally, we obtained 6-MWT-derived variables, which included 6-MWT speed (6-MWS), calculated by dividing the distance by the total walking time (360 s) [16]; distance–saturation product (DSP), calculated from the distance walked versus final SpO_2_ [17]; and absolute desaturation (AD), defined as the difference between pre-SpO_2_ minus post-6-MWT SpO_2_. Predicted D6-MWT values were based on a reference model for healthy Brazilian individuals [18].

#### 2.3.3. Cycle Ergometer CPET

For CPET, patients underwent an incremental ramp protocol using a cycle ergometer (CG—04, Inbramed, Porto Alegre, RS, Brazil), with the load progressively increasing every minute until exhaustion, following international recommendations [19]. The seat height and arm position were adjusted to achieve maximal efficiency for each patient. An oronasal mask was used to measure the levels of expiratory gases captured and continuously registered for each breath (Quark PFT; COSMED, Rome, Italy). Additionally, continuous HR was recorded using an electrocardiogram (Welch Allyn Cardioperfect, Skaneateles Falls, NY, USA) and pulse oximetry (Nonin Medical Ipod, Minneapolis, MN, USA). Finally, noninvasive blood pressure measurements were obtained every 2 min (Premium Hospital Sphygmomanometer; Shanghai, China).

The maximum predicted oxygen consumption (V’ O_2_), patient age, dyspnea level during exercise, and exercise capacity were used to provide an incremental period of 8–12 min according to the American Thoracic Society (ATS) recommendations [19]. Expiratory gas measurements were recorded for the analysis of ventilatory and cardiovascular variables: carbon dioxide production (V’ CO_2_), end-tidal expiratory carbon dioxide pressure (PetCO_2_) absolute (V’ E máx) and predicted (V’ E% pred) maximal minute ventilation, absolute (V’ O_2_ max) and predicted (V’ O_2_% pred) oxygen consumption, and minute ventilation/carbon dioxide production (V’ E/V’ CO_2_). The anaerobic threshold was estimated using non-invasive gas exchange methods.

For the 6-MWT and CPET, the predicted values for each patient were considered according to anthropometric and demographic variables.

### 2.4. Statistical Analysis

Spirometric, 6-MWT, and CPET variables were analyzed, with the percentage of predicted values considered to account for age differences between the groups. Normal distribution was assessed using the Kolmogorov–Smirnov test. Parametric or non-parametric Mann–Whitney U tests were used. Intergroup comparisons were conducted using analysis of variance (ANOVA) or Kruskal–Wallis tests, with Tukey’s or Mann–Whitney tests for paired comparisons. Pearson’s correlation analysis was performed considering both groups as single samples. Statistical significance was set at *p* < 0.05. All analyses were conducted using SPSS 20.0 for Windows (IBM Corp. Released 2011, Armonk, NY, USA).

## 3. Results

We collected data from 20 individuals, comprising 10 patients with post-acute COVID-19 infection patients and 10 with restrictive lung disease, as shown in Figure 1. Of the 55 patients post-SARS-CoV-2 infection initially eligible for the rehabilitation program after discharge from the hospital, 10 were enrolled in the final analysis.

The demographic and baseline spirometric data of the post-acute COVID-19 group are presented in Table 1. Of the eight patients with severe initial infection, seven were placed on invasive mechanical ventilation during their hospital stay (#1, 3, 4, 5, 7, 8, 9), two were placed on noninvasive mechanical ventilation (#4, 10), three (#2, 5 and 6) had a history of smoking, and none had a history of lung diseases. by cardiologist previously followed up with six patients because of high blood pressure without further cardiac complications or a history of previous cardiac events.

The group of patients with chronic respiratory disorder was composed of five males. They had various RLD conditions, including interstitial lung diseases secondary to connective tissue disease (*n* =2), interstitial lung abnormalities (*n* = 2), unclassified interstitial lung disease (*n* = 2), cryptogenic organizing lung disease (*n* = 1), interstitial lung disease secondary to gastroesophageal reflux disease (*n* = 1), desquamative interstitial lung disease (*n* = 1), and idiopathic pulmonary fibrosis (*n* = 1). Three patients reported a history of smoking. Overall, the patients had a mean age of 63.5 + 11.4 years, mean BMI values of 32.0 + 5.7 kg/m^2^, mean FVC (% predicted) values of 76.5 + 16.8 and FEV1/FVC mean values of 95.5 + 11.5%. Finally, among the common comparable data (sex, age, BMI, FVC, and FEV1/FVC ratio), only age significantly differed between the RLD and post-acute COVID-19 groups (Table 2).

Table 3 presents data on both groups’ submaximal (6-MWT) and maximal (CPET) effort capacity with respect to submaximal (6MWT) and maximal (CPET) efforts.

For variables that did not present significant differences between the groups, that is, spirometric 6-MWT and CPET variables, we proceeded with a correlation analysis between resting variables (spirometric) and those of submaximal (6-MWT) and maximal (CPET) effort capacities, considering both groups as unique samples. Table 4 presents the Pearson correlation analysis between the resting, submaximal, and maximal effort variables, which showed significant correlations.

## 4. Discussion

Our study attempts to analyze the submaximal and maximal effort capacities of individuals with post-acute COVID-19 infection and those with other chronic lung conditions paired with pulmonary function tests. The literature indicates that the most prevalent impairments in lung function among individuals with post-acute COVID-19 infection are altered diffusion capacity [20] and the development of a pulmonary restrictive pattern [9]. Identifying exercise impairments in these patients is crucial [21], as even months following hospital discharge, patients with post-acute COVID-19 infection have been shown to experience aerobic capacity deficits due to physical deconditioning [22] and peripheral muscle mass impairment [12]. Thus, it is important to describe the extent of pulmonary impairment, which may guide supplemental oxygen introduction during exercise, and monitor eventual functional impairments during follow-up, targeting appropriate interventions in physiological systems (respiratory, cardiovascular, or musculoskeletal). Concerning the initial pulmonary infection with SARS-CoV-2, it is reasonable to assume that ventilation-perfusion mismatches contribute to dyspnea and hypoxemic events, ultimately affecting the overall functional capacity. The maintenance of hypoxemic conditions may lead to musculoskeletal impairment, as observed in chronic respiratory diseases.

Our initial group of patients with SARS-CoV-2 infection predominantly comprised individuals with severe/critical infections [23], with a significant proportion presenting overweight/obesity and at least one previous comorbidity. Notably, 22% of the initially referred patients with post-acute COVID-19 infection had underlying lung diseases. In our study sample, age was the only demographic difference between the groups. The RLD group was older, as expected, given that this condition is more prevalent in older individuals [24]. The observation of this fact helped guide our subsequent analyses; we referred to the variables as percentages of predicted values, thus seeking to normalize the analyses concerning age. Therefore, age-related differences between groups should be considered when interpreting exercise performance. Older individuals experience progressive declines in maximal oxygen uptake (V’ O_2_ max), muscle mass, and cardiovascular efficiency, all of which independently contribute to reduced functional capacity [25]. When spirometry-matched comparisons were used, the older age of the RLD group may have influenced the CPET outcomes, particularly those related to ventilatory efficiency and maximal heart rate responses.

After this consideration and adjusted analyses, the differences in submaximal (6-MWT) and maximal efforts (CPET) between the groups revealed similar performances, except for DDP and AD for the 6-MWT, and HR and V’ E max for the CPET. Concerning submaximal effort analyses, we assumed that the RLD group, which walked a shorter distance and had more pronounced desaturation during exercise, had significant differences in DDP and AD in the 6-MWT. Even if it was not significantly different, the RLD group performed a shorter absolute distance than the post-COVID-19 group (406.3 ± 85.3 vs. 474.7 ± 86.8 m, respectively). The performance presented in the 6-MWT for both groups was similar to that in a previous study concerning post-COVID-19 patients [26]. and those with RLD pith similar FVC values [27].

In the initial CPET analysis, the comparison of absolute variables between post-COVID-19 and RLD groups exhibited significant differences in absolute V’ O_2_ max (64.7 ± 14.5 vs. 12.3 + 2.6 mL/kg/min, respectively) and V’ E max (11.25 + 15.9 vs. 39.2 ± 12.0 L/min, respectively). We noticed that comparing absolute V’ O_2_ max with typical predicted values revealed reduced values in all post-acute COVID-19 individuals [28], consistent with severe acute respiratory syndrome data [29]. When normalized using the expected values, V’ O_2_ max did not differ significantly between the groups, suggesting that the significant age difference played a major role. Similarly, age differences likely explain the discrepancies found in the heart rate variables. In contrast, V’ E max (% predicted) showed significant differences influenced not only by age but also by chronic lung function impairment and compromised ventilatory efficiency in the RLD group.

Since the study groups were matched based on pulmonary function, no significant differences in 6-MWT (% predicted) were observed between them. By plotting the distribution data of both groups and analyzing the correlation analysis concerning spirometric variables and those of the 6-MWT and CPET (Figure 2), we found a significant correlation between the D6-MWT (% predicted) and FVC (% predicted). Additionally, when we analyzed the performance according to normal values, i.e., FVC ≥ 80% predicted and D6-MWT ≥ 80% predicted, we noticed that 13 of/20 patients (65% of patients; six post-COVID-19 and seven RLD patients) presented performances below expected either for FVC or D6-MWT. By placing an identity line, we noticed that 6/10 patients with RLD presented with major impairments in functional capacity concerning pulmonary function. In contrast, 7/10 patients with post-COVID-19 infection presented with a relatively more preserved functional capacity with respect to pulmonary function. We hypothesized that, as a transitory condition, functional capacity would be preserved to the detriment of pulmonary function.

The observed correlations between FVC and functional outcomes highlight the crucial role of pulmonary restriction in determining exercise tolerance. The association between a lower FVC and reduced 6-MWT distance suggests that ventilatory constraints significantly affect functional performance. Additionally, the negative correlation between FVC and V’ E max (% predicted) indicates that individuals with greater restriction experience greater ventilatory inefficiency during maximal-effort tasks. This suggests that ventilatory limitations, rather than cardiovascular or muscular limitations, maybe the primary determinant of exercise intolerance in these patients.

Thus, when an initial infection occurs in the respiratory system, this will be the initial location of organic issues [30]. As we evaluated these patients immediately after hospital discharge, the ventilation–perfusion mismatch did not spread its effects except to the skeletal muscles, allowing them to maintain their functional capacities. As we paired the groups using spirometry, the enrolled patients with RLD did not present with severe pulmonary restriction, which probably explains their greater functional capacity impairment. Furthermore, as a chronic condition, skeletal muscle derangements are expected to result from chronic hypoxemia and systemic inflammation [31].

A correlation between FVC and the 6-MWT [32] in the RLD group was expected. We found a similar correlation as previously described by considering post-COVID-19 individuals with similar pulmonary function impairments. In the CPET, significant correlations were found between FVC, V’ E max (% predicted), and V’ E/V’ CO_2_ slopes. As a single group, the restrictive pattern significantly impacted both submaximal and maximal effort capacities [33].

Although the V’ E/V’ CO_2_ slope, a measure of ventilatory efficiency, did not differ between the groups (Table 2) compared to the reference values [34], it showed elevated levels for both, particularly in the RLD group. We expected higher V’ E/V’ CO_2_ levels in the RLD group owing to their older age [35]; however, we failed to find significant differences to find significant differences because of the small sample size. We postulate that ventilatory inefficiency may be attributed to earlier attainment of critical mechanical constraints (restrictive lung pattern), resulting in increased minute ventilation relative to the metabolic demand. Furthermore, pulmonary gas exchange abnormalities and ventilatory inefficiency have been described [10,11] as key mechanisms of dyspnea and exercise intolerance in individuals with interstitial lung diseases. Therefore, it is plausible that individuals with post-acute COVID-19 infection experience a ventilation/perfusion mismatch, leading to increased alveolar dead space and exacerbated ventilatory drive due to exercise-induced hypoxemia, which contributes to increasing minute ventilation relative to CO_2_ production during exercise.

The heterogeneity of the underlying diseases within the RLD group is an important factor to consider. Although all patients exhibited a restrictive spirometric pattern, the pathophysiologies of interstitial lung diseases, cryptogenic organizing pneumonia, and desquamative interstitial pneumonia vary significantly. Some conditions may present with greater degrees of pulmonary fibrosis and diffusion abnormalities, which could differentially affect exercise capacity. Future studies should consider stratifying the analyses based on specific disease etiologies to better understand their functional implications.

Finally, the findings of this study provide valuable insights for designing targeted rehabilitation strategies for post-COVID-19 patients with restrictive pulmonary patterns. Given the observed impairments in both submaximal and maximal exercise capacities, rehabilitation programs should prioritize a multimodal approach, incorporating aerobic training, respiratory muscle strengthening, and peripheral muscle conditioning to improve functional performance. The significant correlations between FVC and 6-MWT distance suggest that interventions aimed at improving lung mechanics and ventilatory efficiency could enhance exercise tolerance. Additionally, considering the potential for ventilation-perfusion mismatches and hypoxemia-related muscle dysfunction, oxygen therapy during exertion and neuromuscular training may be beneficial for selected patients. Given that post-COVID-19 impairments may be transient, periodic reassessments should be integrated into rehabilitation protocols to tailor interventions as pulmonary and musculoskeletal functions recover. Ultimately, these findings reinforce the need for individualized rehabilitation plans that address both pulmonary and systemic impairments in post-COVID-19 patients, optimizing their long-term recovery and quality of life.

### Study Limitations

It is important to acknowledge the small sample size in this study. An estimation of approximately 80 patients per group would be required to achieve a statistical power of at least 70% to detect meaningful differences between post-COVID-19 and chronic restrictive lung disease groups, assuming a moderate effect size (Cohen’s d = 0.5) and a significance level of 0.05, in a two-tailed comparison. This estimation accounts for the variability in functional and pulmonary outcomes (e.g., FVC, 6-MWT distance, and CPET variables) and aims to minimize the risk of Type II errors. Because of the strict inclusion and exclusion criteria applied in this study, the sample size was limited. Since this research is part of a larger project, future analyses will include a larger sample to enhance statistical power and generalizability. Additionally, our study focused specifically on patients referred to our pulmonary rehabilitation program, particularly those with a history of COVID-19 infection and a restrictive lung spirometric pattern, Briefly, as seen in Figure 1, due owing to our strict criteria for final analysis, the final analyzed sample was small. Even so, we presented these preliminary results to present similarities and differences seen in submaximal and maximal efforts and also to present a pattern of correlation between spirometric and these efforts capacities. Regarding CPET data, we did not include oxygen uptake efficiency slope, ventilatory power, or circulatory power, as none of our patients had been diagnosed with cardiac conditions (e.g., congestive heart failure, pulmonary hypertension, or congenital heart disease), which would justify analyzing these parameters.

Despite this limitation, we were able to establish parallels in the overall functional capacity impairment between individuals with post-acute COVID-19 and those with restrictive lung disease.

Given the unique nature of post-acute COVID-19 infection, which may be characterized by transient impairments in pulmonary and musculoskeletal functions rather than a chronic evolution similar to other lung diseases, our study highlights the similarities in functional capacity impairments compared with restrictive lung diseases. Even one year after SARS-CoV-2 infection [36], a significant proportion of post-COVID-19 symptoms are expected to persist, including structural impairments of the lung. The persistence of pulmonary impairments may lead to the development of functional impairments, which may be similar to the functional impairment evolution pathway observed in chronic respiratory lung diseases, in which one of the triggering points is the development of hypoxemia.

Another notable limitation of our study was the lack of chest tomography and carbon monoxide diffusion measurement in individuals with COVID-19. These measures provide valuable information regarding the extent of pulmonary impairment. Similarly, our group of patients with restrictive lung disease consisted of individuals with various restrictive pathologies rather than a single disease entity. Although this approach was chosen to minimize the influence of major comorbidities on functional capacity, it may have introduced heterogeneity into the group. Nonetheless, focusing on restrictive spirometric patterns allowed for a more standardized comparison between the two groups.

## 5. Conclusions

The analyzed post-acute COVID-19 infection patients, exhibiting a spirometric pattern similar to that of chronic restrictive lung diseases, presented a comparable performance in both submaximal and maximal effort tests, with the exception of desaturation product and absolute desaturation and with the exception of most variables normalized using percentages of predicted values of the CPET. The degree of restriction observed in both groups influenced the distance and speed achieved during the 6-MWT, as well as the maximal minute ventilation during maximal effort.

## Figures and Tables

**Figure 1 ijerph-22-00261-f001:**
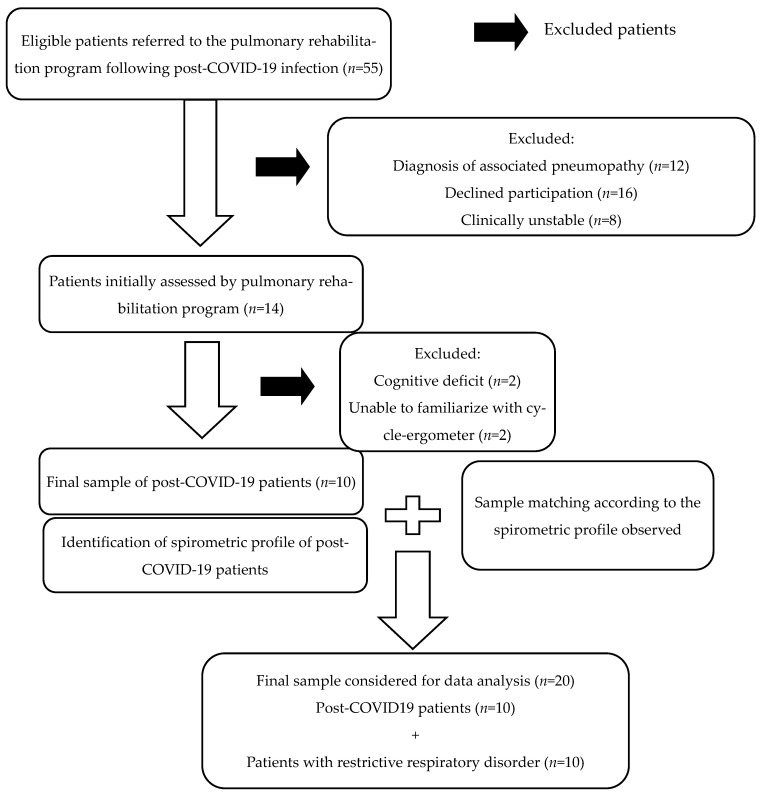
Sample selection flowchart.

**Figure 2 ijerph-22-00261-f002:**
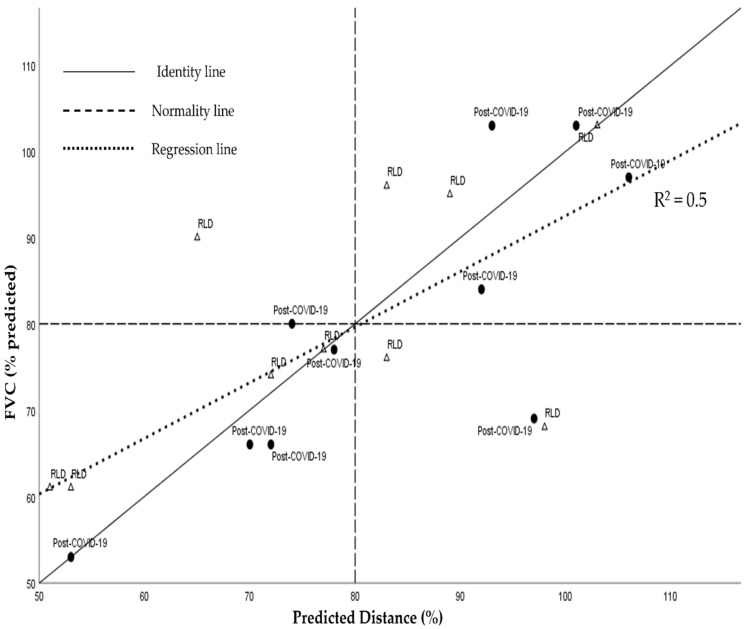
Pearson’s correlation analysis for the whole sample between predicted FVC (% predicted) and Predicted Distance (%). ∆ RLD = respiratory lung disease; ● = Post-COVID-19.

**Table 1 ijerph-22-00261-t001:** Demographic, spirometric, and COVID-19 infection data of evaluated patients.

Patient #	Sex (M/F)	Comorbidities	COVID-19 Infection Severity	Age (Years)	BMI (kg/m^2^)	FEV_1_/FVC (%)	FVC (% Predicted)	Hospital Stay (Days)
1	F	HBP	Severe	47	32.3	91	80	36
2	F	DM	Mild	52	29.0	105	103	10
3	F	HBP/DM	Severe	52	35.2	103	97	30
4	F	HBP + Fibromyalgia	Severe	41	27.6	100	53	30
5	M	HBP	Severe	57	22.0	86	66	60
6	M	HBP + DM	Mild	70	34.9	107	69	10
7	M	DM	Severe	55	29.0	102	84	28
8	M	DM	Severe	40	28.1	99	66	30
9	M	DM + Fibromyalgia	Severe	35	29.1	103	77	42
10	M	DM + HBP	Severe	35	29.0	86	103	18
			Mean ± Std. Dev	49.5 ± 11.0	28.6 ± 4.5	101.1 ± 7.7	78.5 ± 17.0	30.0 ± 15.0

M, male; F, female; BMI, body mass index; FEV_1_/FVC, tiffenneau index; FVC, forced vital capacity; HBP, high blood pressure; DM, type II diabetes mellitus.

**Table 2 ijerph-22-00261-t002:** Demographic and spirometric values of post-COVID-19 and RDL groups.

	Age (Years)		Sex (M/F)		BMI (kg/m^2^)		FVC (% Predicted)		FEV_1_/FVC (%)	
	Post-COVID-19	RLD	Post-COVID-19	RLD	Post-COVID-19	RLD	Post-COVID-19	RLD	Post-COVID-19	RLD
Mean ± Std. dev	49.5 ± 11.0	63.5 ± 11.4	6/4	5/5	28.6 ± 4.5	32.0 ± 5.7	78.5 ± 17.0	76.5 ± 16.8	101.0 ± 7.7	95.5 ± 14.5
*p*-value	0.04		ns		ns		ns		ns	

M, male; F, female; RLD, restrictive lung disease; BMI, body mass index; FVC, forced vital capacity; FEV_1_, forced expiratory volume in the first second; ns, non-significant.

**Table 3 ijerph-22-00261-t003:** 6-MWT and CPET data of the groups.

Submaximal/Maximal Variables	Post-COVID-19	RLD	*p*-Value
D6-MWT (distance: % predicted)	77.3 ± 17.2	83.8 ± 17.0	ns
6-MWS (m/s)	1.12 ± 0.24	1.31 ± 0.24	ns
6-MWW (m × kg)	32,790.6 ± 10,015.9	39,003.8 ± 11,150.3	ns
DDP (m × final SpO_2_)	347.6 ± 66.8	457.6 ± 87.5	0.005
AD for 6-MWT	7.6 ± 5.5	1.9 ± 2.6	0.009
V’ O_2_ max (% predicted)	58.5 ± 15.9	69.4 ± 17.3	ns
HR max (% predicted)	75.9 ± 8.2	87.5 ± 10.1	0.01
V’ E max (%predicted)	51.2 ± 16.0	64.7 ± 14.5	0.01
PetCO_2_ (mmHg)	33.0 ± 7.3	32.0 ± 2.5	ns
V’ E/V’ CO_2_ slope	42.5 + 22.9	33.5 + 4.7	ns

RLD = respiratory lung disease; 6-MWT = 6 min walk test; 6-MWS = speed at 6-MWT; 6-MWW = work at 6-MWT; DDP = distance-desaturation product; AD = absolute desaturation at 6-MWT; V’ O_2_ max = maximum oxygen consumption; HR max = maximum heartbeat; V’ E máx = maximum minute ventilation; PetCO_2_ = end-tidal partial pressure of carbon dioxide; V’ E/V’ CO_2_ slope = minute ventilation/dioxide carbon production; ns = non-significant.

**Table 4 ijerph-22-00261-t004:** Significant correlations among spirometric, 6-MWT, and CPET variables.

		*R* ^2^	*p*-Value
**FVC (% Predicted)**	6-MWT		
	Distance (% predicted)	0.5	0.001
	Speed (m/s)	0.5	0.001
	CPET		
	V’ E max (%)	0.5	<0.05
	V’ E/V’ CO_2_ slope	0.5	<0.05

FVC, Forced Vital Capacity; 6-MWT, 6-min walk test; CPET = Cardiopulmonary exercise test; V’ E max, maximum minute Ventilation; V’ E/V’ CO_2_ slope = minute ventilation/carbon dioxide production.

## Data Availability

The corresponding author stored non-digital data supporting this study at the University Hospital of Brasília. Details on how access to these data are available via email contact with the author.

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
