# Peer review of "Comparative Analysis of Submaximal and Maximal Effort Capacities in Patients Post-COVID-19 and Individuals with Chronic Restrictive Lung Diseases"

_ijerph, 2025, doi:10.3390/ijerph22020261_

Round 1
Reviewer 1 Report
Comments and Suggestions for Authors
The paper „Impairments at submaximal and maximal effort capacities in patients with post-COVID-19 infections and in individuals with chronic respiratory diseases; are they comparable?” covers an interesting study about similarities between post-COVID lung impairment and other restrictive lung diseases. Such comparison could lead eventually to better treatment procedures in post-COVID or long-COVID patients. The most important weakness of the paper is the very small group of patients (10 post-COVID and 10 controls), thus the obtained results are rather of preliminary character and are a good basis for a larger study. However, this has been stated by the authors. Otherwise, the presentation of the population and the results can be ameliorated. The tables should be unified in style and bolding and more demographic data of both populations could be presented. It would be nice to have information about comorbidities of both groups, the respiratory health, smoking habits, cancer etc. Additionally, correct use of English lettering and style of numbers should checked together with typing errors incl. abbreviations. Also the presentation and quality of Fig 2. Should be ameliorated. Taken together, the paper presents in a nice way important findings of preliminary studies about the correlation of lung impairment in post-COVID and other respiratory patients
Comments on the Quality of English LanguageCheck English language for typing errors, avoid letters from other languages.
Author Response
We would like to thank the reviewer for the comments and due to suggestions, we could improve our analysis and this final version. We explored and emphasized even further the findings of our study which could lead to better comprehension, allowing a more precise and targeted treatment procedures in post-COVID-19 patients. Thus, we emphasized the following sentence at introduction section:
“Furthermore, it is crucial to understand functional impairment patterns, which might not be properly evaluated using resting physiological evaluations, and identify the factors contributing to pulmonary-related exercise limitations.”.
We also added the following sentence at discussion section:
“Thus, it is important to describe the extent of pulmonary impairment, which may guide supplemental oxygen introduction during exercise, and monitor eventual functional impairments during follow-up, targeting appropriate interventions in physiological systems (respiratory, cardiovascular, or musculoskeletal)”.
We completely agree with the reviewer with respect to sample size of study as the main weakness of the paper. We had already mentioned at the first version of the paper, but now we cleared even further this limitation. We also stated at this new version the estimated sample size accordingly statistics estimation to better detect meaningful differences, we also highlighted our methodological constraints that culminates in this initial small sample presented, and we have now stated the ‘study limitations’ subtitle:
“It is important to acknowledge the small sample size in this study. An estimation of approximately 80 patients per group would be required to achieve a statistical power of at least 70% to detect meaningful differences between post-COVID-19 and chronic restrictive lung disease groups, assuming a moderate effect size (Cohen’s d = 0.5) and a significance level of 0.05, in a two-tailed comparison. This estimation accounts for the variability in functional and pulmonary outcomes (e.g., FVC, 6-MWT distance, and CPET variables) and aims to minimize the risk of Type II errors. Because of the strict inclusion and exclusion criteria applied in this study, the sample size was limited. Since this research is part of a larger project, future analyses will include a larger sample to enhance statistical power and generalizability. Additionally, our study focused specifically on patients referred to our pulmonary rehabilitation program, particularly those with a history of COVID-19 infection and a restrictive lung spirometric pattern, Briefly, as seen in the Figure 1, due owing to our strict criteria for final analysis, the final analyzed sample was small. Even so we presented these preliminary results to present similarities and differences seen in submaximal and maximal efforts, and also to present pattern of correlation between spirometric and these efforts capacities.
At this new version we have also improved the presentation of post-COVID-19 group, by adding the following sentences at ‘Results’ section:
“The demographic and baseline spirometric data of the post-acute COVID-19 group are presented in Table 1. Of the eight patients with severe initial infection, seven were placed on invasive mechanical ventilation during their hospital stay (#1, 3, 4, 5, 7, 8, 9), two were placed on noninvasive mechanical ventilation (#4, 10), three (#2, 5 and 6) had a history of smoking, and none had a history of lung diseases. by cardiologist previously followed up with six patients because of high blood pressure without further cardiac complications or a history of previous cardiac events”.
We have changed the table 1 for this new version, which presents data of each patient individually:
Table 1. Demographic, spirometric, and COVID-19 infection data of evaluated patients.
|
Patient # |
Sex (M/F) |
Comorbidities |
COVID-19 infection severity |
Age (years) |
BMI (kg/m2) |
FEV1/ FVC (%) |
FVC (% predicted) |
Hospital Stay (days) |
|
|
1 |
F |
HBP |
Severe |
47 |
32.3 |
91 |
80 |
36 |
|
|
2 |
F |
DM |
Mild |
52 |
29.0 |
105 |
103 |
10 |
|
|
3 |
F |
HBP/DM |
Severe |
52 |
35.2 |
103 |
97 |
30 |
|
|
4 |
F |
HBP+Fibromyalgia |
Severe |
41 |
27.6 |
100 |
53 |
30 |
|
|
5 |
M |
HBP |
Severe |
57 |
22.0 |
86 |
66 |
60 |
|
|
6 |
M |
HBP+DM |
Mild |
70 |
34.9 |
107 |
69 |
10 |
|
|
7 |
M |
DM |
Severe |
55 |
29.0 |
102 |
84 |
28 |
|
|
8 |
M |
DM |
Severe |
40 |
28.1 |
99 |
66 |
30 |
|
|
9 |
M |
DM+Fibromyalgia |
Severe |
35 |
29.1 |
103 |
77 |
42 |
|
|
10 |
M |
DM+HBP |
Severe |
35 |
29.0 |
86 |
103 |
18 |
|
|
|
|
|
|
|
|
|
|
|
|
|
|
|
|
Mean+Std. Dev |
49.5+ 11.0 |
28.6+ 4.5 |
101.1+ 7.7 |
78.5+ 17.0 |
30.0+ 15.0 |
|
Finally, we also agree with the reviewer concerning the presentation of figure 2, and now we have re-built it to improve visualization and comprehension, as follows:
RLD = respiratory lung diseases
We appreciate all suggestions made by the reviewer, which allowed us to better present and improve the quality of our article.

Reviewer 2 Report
Comments and Suggestions for Authors
This manuscript addresses the important topic of exercise tolerance impairment following COVID-19 infection, which is interesting. However, the study design has several significant flaws that need to be addressed to strengthen the findings' validity.
1. The conclusion suggests that the two patient groups have comparable performance in 6MWD and CPET. However, this finding is questionable given the small sample size. The lack of a statistically significant difference between the two groups is likely due to insufficient statistical power. I recommend that the study design include an estimation of the required sample size to ensure adequate power for detecting meaningful differences.
2. Table 1 presents incomplete baseline clinical characteristics. More detailed information about the patients enrolled in this study should be provided, including comorbidities. Additionally, complete PFTs parameters (e.g., TLC, FRC, MVV) should be included.
3. CPET is an invaluable tool in healthcare; however, its full potential is not reflected in this study, as the authors focused on overly simplistic and limited parameters. More comprehensive parameters, such as PETCO2, oxygen pulse, OUES, ventilatory power, and circulatory power, would be highly valuable for exploring exercise patterns and cardiopulmonary function in patients with impaired exercise capacity.
4. The correlation analysis between resting, submaximal, and maximal effort variables is well-established in previous studies. Repeating these analyses does not provide new insights. The manuscript should focus on identifying other innovative aspects.
Author Response
We would like to thank the reviewer for the comments and due to suggestions, we could improve our analysis and this final version. We explored and emphasized even further the findings of our study which could lead to better comprehension, allowing a more precise and targeted treatment procedures in post-COVID-19 patients.
- We completely agree with the reviewer with respect to sample size of study as the main weakness of the paper. We had already mentioned at the first version of the paper, but now we cleared even further this limitation. We also stated at this new version the estimated sample size accordingly statistics estimation to better detect meaningful differences, we also highlighted our methodological constraints that culminates in this initial small sample presented, and we have now stated the ‘study limitations’ subtitle:
“It is important to acknowledge the small sample size in this study. An estimation of approximately 80 patients per group would be required to achieve a statistical power of at least 70% to detect meaningful differences between the post-COVID-19 and chronic restrictive lung disease groups, assuming a moderate effect size (Cohen’s d = 0.5) and a significance level of 0.05, in a two-tailed comparison. This estimation accounts for the variability in functional and pulmonary outcomes (e.g., FVC, 6-MWT distance, and CPET variables) and aims to minimize the risk of Type II errors. However, due to limited final sample size owing to the strict inclusion and exclusion criteria applied in this study. As this study is a part of a larger research project, we anticipate increasing the sample size in future analyses to enhance the statistical power. Additionally, our study focused specifically on patients referred to our pulmonary rehabilitation program, particularly those with a history of COVID-19 infection and a restrictive lung spirometric pattern, Briefly, as seen in the Figure 1, due owing to our strict criteria for final analysis, the final analyzed sample was small. Even so we presented these preliminary results to present similarities and differences seen in submaximal and maximal efforts, and also to present pattern of correlation between spirometric and these efforts capacities.”.
The conclusion section was also modified as follows:
“The analyzed post-acute COVID-19 infection patients, exhibiting a spirometric pattern similar to that of chronic restrictive lung diseases, presented a comparable performance in both submaximal and maximal effort tests, with exception of desaturation product and absolute desaturation and with exception of most variables normalized using pentages of predicted values of theat CPET. The degree of restriction observed in both groups influenced the distance and speed achieved during the 6- MWT, as well as the maximal minute ventilation during maximal effort”.
- At this new version we have also improved the presentation of post-COVID-19 group, by adding the following sentences at ‘Results’ section:
“The demographic and baseline spirometric data of the post-acute COVID-19 group are presented in Table 1. Of the eight patients with severe initial infection, seven were placed on invasive mechanical ventilation during their hospital stay (#1, 3, 4, 5, 7, 8, 9), two were placed on noninvasive mechanical ventilation (#4, 10), three (#2, 5 and 6) had a history of smoking, and none had a history of lung diseases. by cardiologist previously followed up with six patients because of high blood pressure without further cardiac complications or a history of previous cardiac events”.
We have changed the table 1 for this new version, which presents data of each patient individually:
Table 1. Demographic, spirometric, and COVID-19 infection data of evaluated patients.
|
Patient # |
Sex (M/F) |
Comorbidities |
COVID-19 infection severity |
Age (years) |
BMI (kg/m2) |
FEV1/ FVC (%) |
FVC (% predicted) |
Hospital Stay (days) |
|
|
1 |
F |
HBP |
Severe |
47 |
32.3 |
91 |
80 |
36 |
|
|
2 |
F |
DM |
Mild |
52 |
29.0 |
105 |
103 |
10 |
|
|
3 |
F |
HBP/DM |
Severe |
52 |
35.2 |
103 |
97 |
30 |
|
|
4 |
F |
HBP+Fibromyalgia |
Severe |
41 |
27.6 |
100 |
53 |
30 |
|
|
5 |
M |
HBP |
Severe |
57 |
22.0 |
86 |
66 |
60 |
|
|
6 |
M |
HBP+DM |
Mild |
70 |
34.9 |
107 |
69 |
10 |
|
|
7 |
M |
DM |
Severe |
55 |
29.0 |
102 |
84 |
28 |
|
|
8 |
M |
DM |
Severe |
40 |
28.1 |
99 |
66 |
30 |
|
|
9 |
M |
DM+Fibromyalgia |
Severe |
35 |
29.1 |
103 |
77 |
42 |
|
|
10 |
M |
DM+HBP |
Severe |
35 |
29.0 |
86 |
103 |
18 |
|
|
|
|
|
|
|
|
|
|
|
|
|
|
|
|
Mean+Std. Dev |
49.5+ 11.0 |
28.6+ 4.5 |
101.1+ 7.7 |
78.5+ 17.0 |
30.0+ 15.0 |
|
- We completely agree with the reviewer about CPET data. At this new version we stated the following sentences with respect to our initially limited use of these information.:
“Regarding CPET data, we did not include oxygen uptake efficiency slope, ventilatory power, or circulatory power, as none of our patients had diagnosed cardiac conditions (e.g., congestive heart failure, pulmonary hypertension, or congenital heart disease), which would justify analyzing these parameters”.
- Concerning the correlation data presented, we modified it at this ‘discussion’ section as follows:
“Since the study groups were matched based on pulmonary function, no significant differences in 6-MWT (% predicted) were observed between them. By ploting the distribution data of both groups, and analyzing on the correlation analysis concerning spirometric variables and those of the 6-MWT and CPET (Figure 2), we found a significant correlation between the D6MWT (% predicted) and FVC (% predicted). Additionally, when we analyzed the performance according to normal values, i.e., FVC > 80% predicted and D6-MWT > 80% predicted, we noticed that 13 of /20 patients (65% of patients; six post-COVID-19 and seven RLD patients) presented performances below expected either for FVC or D6-MWT. By placing an identity line, we noticed that 6/10 patients with RLD presented with major impairments in functional capacity concerning pulmonary function. In contrast, 7/10 patients with post-COVID-19 infection presented with a relatively more preserved functional capacity with respect to pulmonary function. We hypothesized that, as a transitory condition, functional capacity would be preserved to the detriment of pulmonary function.”
Finally, we also agree with the reviewer concerning the presentation of figure 2, and now we have re-built it to improve visualization and comprehension, as follows:
RLD = respiratory lung diseases
We appreciate all suggestions made by the reviewer, which allowed us to better present and improve the quality of our article.

Reviewer 3 Report
Comments and Suggestions for Authors
The authors presented the results obtained on a very small sample. In human studies, the sample size of the participant group cannot be less than 20 subjects. If the sample is small, then the homogeneity of the sample, including disease state and gender, must be taken into account. Furthermore, the authors in this study analyzed groups of patients with diseases of completely different etiologies without the pathophysiological background and assumptions taken into account. The aim and results of the study were not to examine whether they are comparable. Therefore, the title is not related to the manuscript and should be corrected. Also, the results are not clear and Figure 2 needs to be improved. The discussion and conclusion are superficial and shallow. Overall, this study lacks a proper methodological basis and I suggest a major revision.
Author Response
- We agree with the reviewer with respect to sample size of study as the main weakness of the paper. We had already mentioned at the first version of the paper, but now we cleared even further this limitation. We also stated at this new version the estimated sample size accordingly statistics estimation to better detect meaningful differences, we also highlighted our methodological constraints that culminates in this initial small sample presented, and we have now stated the ‘study limitations’ subtitle:
“It is important to acknowledge the small sample size in this study. An estimation of approximately 80 patients per group would be required to achieve a statistical power of at least 70% to detect meaningful differences between post-COVID-19 and chronic restrictive lung disease groups, assuming a moderate effect size (Cohen’s d = 0.5) and a significance level of 0.05, in a two-tailed comparison. This estimation accounts for the variability in functional and pulmonary outcomes (e.g., FVC, 6-MWT distance, and CPET variables) and aims to minimize the risk of Type II errors. Because of the strict inclusion and exclusion criteria applied in this study, the sample size was limited. Since this research is part of a larger project, future analyses will include a larger sample to enhance statistical power and generalizability. Additionally, our study focused specifically on patients referred to our pulmonary rehabilitation program, particularly those with a history of COVID-19 infection and a restrictive lung spirometric pattern, Briefly, as seen in the Figure 1, due owing to our strict criteria for final analysis, the final analyzed sample was small. Even so we presented these preliminary results to present similarities and differences seen in submaximal and maximal efforts, and also to present pattern of correlation between spirometric and these efforts capacities.”
- With respect to ‘discussion’ and ‘conclusion’ sections, we do not agree that we have presented a shallow and superficial argumentation. Even so, we have modified as follows to avoid misinterpretations:
“4. Discussion
Our study attempts to analyze the submaximal and maximal effort capacities of individuals with post-acute COVID-19 infection and those with other chronic lung conditions paired with pulmonary function tests. The literature indicates that the most prevalent impairments in lung function among individuals with post-acute COVID-19 infection are altered diffusion capacity [20] and the development of a pulmonary restrictive pattern [9]. Identifying exercise impairments in these patients is crucial [21], as even months following hospital discharge, patients with post-acute COVID-19 infection have been shown to experience aerobic capacity deficits due to physical deconditioning [22] and peripheral muscle mass impairment [12]. Thus, it is important to describe the extent of pulmonary impairment, which may guide supplemental oxygen introduction during exercise, and monitor eventual functional impairments during follow-up, targeting appropriate interventions in physiological systems (respiratory, cardiovascular, or musculoskeletal). Concerning the initial pulmonary infection with SARS-CoV2, it is reasonable to assume that ventilation-perfusion mismatches contribute to dyspnea and hypoxemic events, ultimately affecting the overall functional capacity. The maintenance of hypoxemic conditions may lead to musculoskeletal impairment, as observed in chronic respiratory diseases.
Our initial group of patients with SARS-CoV-2 infection predominantly comprised individuals with severe/critical infections [23], with a significant proportion presenting overweight/obesity and at least one previous comorbidity. Notably, 22% of the initially referred patients with post-acute COVID-19 infection had underlying lung diseases. In our study sample, age was the only demographic difference between the groups. The RLD group was older, as expected, given that this condition is more prevalent in older individuals [24]. The observation of this fact helped guide our subsequent analyses; we referred to the variables as percentages of predicted values, thus seeking to normalize the analyses concerning age. Therefore, age-related differences between groups should be considered when interpreting exercise performance. Older individuals experience progressive declines in maximal oxygen uptake (V’O2max), muscle mass, and cardiovascular efficiency, all of which independently contribute to reduced functional capacity [25]. When spirometry-matched comparisons were used, the older age of the RLD group may have influenced the CPET outcomes, particularly those related to ventilatory efficiency and maximal heart rate responses.
After this consideration and adjusted analyses, the differences in submaximal (6 MWT) and maximal efforts (CPET) between the groups revealed similar performances, except for DDP and AD for the 6-MWT, and HR and V’Emáx for the CPET. Concerning submaximal effort analyses, we assumed that the RLD group, which walked a shorter distance and had more pronounced desaturation during exercise, had significant differences in DDP and AD in the 6- MWT. Even if it was not significantly different, the RLD group performed a shorter absolute distance than the post-COVID-19 group (406.3+85.3 vs 474.7+86.8 m, respectively). The performance presented in the 6- MWT for both groups was similar to that in a previous study concerning post-COVID-19 patients [26]. and those with RLD pith similar FVC values [27].
In the initial CPET analysis, the comparison of absolute variables between post-COVID-19 and RLD groups exhibited significant differences in absolute V’O2max (64.7+14.5 vs. 12.3+2.6 mL/kg/min, respectively) and V’Emax (11.25+15.9 vs. 39.2+12.0 L/min, respectively). We noticed that comparing absolute V’O2max with typical predicted values revealed reduced values in all post-acute COVID-19 individuals [28], consistent with severe acute respiratory syndrome data [29]. When normalized using the expected values, V’O2max did not differ significantly between the groups, suggesting that the significant age difference played a major role. Similarly, age differences likely explain the discrepancies found in the heart rate variables. In contrast, V’Emax (% predicted) showed significant differences influenced not only by age but also by chronic lung function impairment and compromised ventilatory efficiency in the RLD group.
Since the study groups were matched based on pulmonary function, no significant differences in 6-MWT (% predicted) were observed between them. By ploting the distribution data of both groups, and analyzing on the correlation analysis concerning spirometric variables and those of the 6-MWT and CPET (Figure 2), we found a significant correlation between the D6MWT (% predicted) and FVC (% predicted). Additionally, when we analyzed the performance according to normal values, i.e., FVC > 80% predicted and D6-MWT > 80% predicted, we noticed that 13 of /20 patients (65% of patients; six post-COVID-19 and seven RLD patients) presented performances below expected either for FVC or D6-MWT. By placing an identity line, we noticed that 6/10 patients with RLD presented with major impairments in functional capacity concerning pulmonary function. In contrast, 7/10 patients with post-COVID-19 infection presented with a relatively more preserved functional capacity with respect to pulmonary function. We hypothesized that, as a transitory condition, functional capacity would be preserved to the detriment of pulmonary function.
RLD = respiratory lung diseases
The observed correlations between FVC and functional outcomes highlight the crucial role of pulmonary restriction in determining exercise tolerance. The association between a lower FVC and reduced 6-MWT distance suggests that ventilatory constraints significantly affect functional performance. Additionally, the negative correlation between FVC and V’Emax (% predicted) indicates that individuals with greater restriction experience greater ventilatory inefficiency during maximal-effort tasks. This suggests that ventilatory limitations, rather than cardiovascular or muscular limitations, may be the primary determinant of exercise intolerance in these patients.
Thus, when an initial infection occurs in the respiratory system, this will be the initial location of organic issues [30]. As we evaluated these patients immediately after hospital discharge, the ventilation-perfusion mismatch did not spread its effects affect to skeletal muscles, allowing them to maintain their functional capacities. As we paired the groups using spirometry, the enrolled patients with RLD did not present with severe pulmonary restriction, which probably explains their greater functional capacity impairment. Furthermore, as a chronic condition, skeletal muscle derangements are expected to result from chronic hypoxemia and systemic inflammation [31].
A correlation between FVC and the 6-MWT [32] in the RLD group was expected. We found a similar correlation as previously described by considering post-COVID-19 individuals with similar pulmonary function impairments. In the CPET, significant correlations were found between FVC, V’Emax (% predicted), and V’E/V’CO2 slopes. As a single group, the restrictive pattern significantly impacted both submaximal and maximal effort capacities [33].
Although the V’E/V’CO2 slope, a measure of ventilatory efficiency, did not differ between the groups (Table 2) compared to the reference values [34], it showed elevated levels for both, particularly in the RLD group. We expected higher V’E/V’CO2 levels in the RLD group owing to their older age [35]; however, we failed to find significant differences to find significant differences because of the small sample size. We postulate that ventilatory inefficiency may be attributed to earlier attainment of critical mechanical constraints (restrictive lung pattern), resulting in increased minute ventilation relative to the metabolic demand. Furthermore, pulmonary gas exchange abnormalities and ventilatory inefficiency have been described [10,11] as key mechanisms of dyspnea and exercise intolerance in individuals with interstitial lung diseases. Therefore, it is plausible that individuals with post-acute COVID-19 infection experience a ventilation/perfusion mismatch, leading to increased alveolar dead space and exacerbated ventilatory drive due to exercise-induced hypoxemia, which contributes to increase minute ventilation relative to CO2 production during exercise.
The heterogeneity of the underlying diseases within the RLD group is an important factor to consider. Although all patients exhibited a restrictive spirometric pattern, the pathophysiologies of interstitial lung diseases, cryptogenic organizing pneumonia, and desquamative interstitial pneumonia vary significantly. Some conditions may present with greater degrees of pulmonary fibrosis and diffusion abnormalities, which could differentially affect exercise capacity. Future studies should consider stratifying the analyses based on specific disease etiologies to better understand their functional implications.
Finally, the findings of this study provide valuable insights for designing targeted rehabilitation strategies for post-COVID-19 patients with restrictive pulmonary patterns. Given the observed impairments in both submaximal and maximal exercise capacities, rehabilitation programs should prioritize a multimodal approach, incorporating aerobic training, respiratory muscle strengthening, and peripheral muscle conditioning to improve functional performance. The significant correlations between FVC and 6MWT distance suggest that interventions aimed at improving lung mechanics and ventilatory efficiency could enhance exercise tolerance. Additionally, considering the potential for ventilation-perfusion mismatches and hypoxemia-related muscle dysfunction, oxygen therapy during exertion and neuromuscular training may be beneficial for selected patients. Given that post-COVID-19 impairments may be transient, periodic reassessments should be integrated into rehabilitation protocols to tailor interventions as pulmonary and musculoskeletal functions recover. Ultimately, these findings reinforce the need for individualized rehabilitation plans that address both pulmonary and systemic impairments in post-COVID-19 patients, optimizing their long-term recovery and quality of life.
4.1. Study Limitations
It is important to acknowledge the small sample size in this study. An estimation of approximately 80 patients per group would be required to achieve a statistical power of at least 70% to detect meaningful differences between post-COVID-19 and chronic restrictive lung disease groups, assuming a moderate effect size (Cohen’s d = 0.5) and a significance level of 0.05, in a two-tailed comparison. This estimation accounts for the variability in functional and pulmonary outcomes (e.g., FVC, 6-MWT distance, and CPET variables) and aims to minimize the risk of Type II errors. Because of the strict inclusion and exclusion criteria applied in this study, the sample size was limited. Since this research is part of a larger project, future analyses will include a larger sample to enhance statistical power and generalizability. Additionally, our study focused specifically on patients referred to our pulmonary rehabilitation program, particularly those with a history of COVID-19 infection and a restrictive lung spirometric pattern, Briefly, as seen in the Figure 1, due owing to our strict criteria for final analysis, the final analyzed sample was small. Even so we presented these preliminary results to present similarities and differences seen in submaximal and maximal efforts, and also to present pattern of correlation between spirometric and these efforts capacities. Regarding CPET data, we did not include oxygen uptake efficiency slope, ventilatory power, or circulatory power, as none of our patients had diagnosed cardiac conditions (e.g., congestive heart failure, pulmonary hypertension, or congenital heart disease), which would justify analyzing these parameters.
Despite this limitation, we were able to establish parallels in the overall functional capacity impairment between individuals with post-acute COVID-19 and those with restrictive lung disease.
Given the unique nature of post-acute COVID-19 infection, which may be characterized by transient impairments in pulmonary and musculoskeletal functions rather than a chronic evolution similar to other lung diseases, our study highlights the similarities in functional capacity impairments compared with restrictive lung diseases. , Even one year after SARS-CoV-2 infection [36], a significant proportion of post-COVID-19 symptoms are expected to persist, including structural impairments of the lung. The persistence of pulmonary impairments may lead to the development of functional impairments, which may be similar to the functional impairment evolution pathway observed in chronic respiratory lung diseases, in which one of the triggering points is the development of hypoxemia.
Another notable limitation of our study was the lack of chest tomography and carbon monoxide diffusion measurement in individuals with COVID-19. These measures provide valuable information regarding the extent of pulmonary impairment. Similarly, our group of patients with restrictive lung disease consisted of individuals with various restrictive pathologies rather than a single disease entity. Although this approach was chosen to minimize the influence of major comorbidities on functional capacity, it may have introduced heterogeneity into the group. Nonetheless, focusing on restrictive spirometric patterns allowed for a more standardized comparison between the two groups.
- Conclusions
The analyzed post-acute COVID-19 infection patients, exhibiting a spirometric pattern similar to that of chronic restrictive lung diseases, presented a comparable performance in both submaximal and maximal effort tests, with exception of desaturation product and absolute desaturation and with exception of most variables normalized using percentages of predicted values of the CPET. The degree of restriction observed in both groups influenced the distance and speed achieved during the 6- MWT, as well as the maximal minute ventilation during maximal effort.”
As presented, we improved the quality of figure 2 to allow a better interpretation.
We have also changed the title to improve the relation with the manuscript, as follows:
“Comparative Analysis of Submaximal and Maximal Effort Capacities in Patients Post-COVID-19 and Individuals With Chronic Restrictive Lung Diseases”

Round 2
Reviewer 2 Report
Comments and Suggestions for Authors
This version is much better. I have no further comments.
Reviewer 3 Report
Comments and Suggestions for Authors
I recommend paper for publication in the present form.